# The Effects of Lifestyle on COVID-19 Vaccine Hesitancy in the United States: An Analysis of Market Segmentation

**DOI:** 10.3390/ijerph19137732

**Published:** 2022-06-24

**Authors:** Esra Ozdenerol, Jacob Seboly

**Affiliations:** 1Spatial Analysis and Geographic Education Laboratory, Department of Earth Sciences, University of Memphis, Memphis, TN 38152, USA; 2Department of Geosciences, Mississippi State University, Starkville, MS 39762, USA; jds1565@msstate.edu

**Keywords:** lifestyle traits, consumer segmentation, COVID-19 vaccination, vaccination intervention, geographic information systems

## Abstract

The aim of this study is to correlate lifestyle characteristics to COVID-19 vaccination rates at the U.S. County level and provide where and when COVID-19 vaccination impacted different households. We grouped counties by their dominant LifeMode, and the mean vaccination rates per LifeMode are calculated. A 95% confidence interval for both the mean and median vaccination rate for each LifeMode is generated. The limits of this interval were compared to the nationwide statistics to determine whether each LifeMode’s vaccine uptake differs significantly from the nationwide average. We used Environmental Systems Research Institute Inc. (ESRI) Tapestry LifeModes data that are collected at the U.S. household level through geodemographic segmentation typically used for marketing purposes. High risk Lifestyle segments and their locations are clearly the areas in the U.S. where the public might benefit from a COVID-19 vaccine. We then used logistic regression analysis to predict vaccination rates using ESRI’s tapestry segmentation and other demographic variables. Our findings demonstrate that vaccine uptake appears to be highest in the urban corridors of the Northeast and the West Coast and in the retirement communities of Arizona and Florida and lowest in the rural areas of the Great Plains and Southeast. Looking closely at other parts of the West such as the Dakotas and Montana, counties that contain Native American reservations have higher vaccination rates. Racial/ethnic minorities also adopt the vaccine at higher rates. The most effective predictor of vaccination hesitancy was Republican voting habits, with Republican counties less likely to take the vaccine. The other predictors in order of importance were college education, minority race/ethnicity, median income, and median age. Our approach correlating lifestyle characteristics to COVID-19 vaccination rate at the U.S. County level provided unique insights into where and when COVID-19 vaccination impacted different households. The results suggest that prevention and control policies can be implemented to those specific households.

## 1. Introduction

Since the U.S. Food and Drug Administration (FDA) authorized the first COVID-19 vaccines, more than one hundred million people in the U.S. have been vaccinated. The initial U.S. COVID-19 vaccination effort potentially reduced COVID-19 infections, hospitalizations, and deaths in the United States [1]. However, many people remain hesitant to receive the vaccine. Our research and market segmentation tools can help identify the lifestyle traits associated with vaccine hesitancy, enabling vaccine uptake to be predicted at local (e.g., Zip Code, census tract, block group) levels [2]. Our targeted approach provides key stake holders to use these tools on their interventions and focus on these specific households, who are exhibiting patterns of vaccination hesitancy but are likely to see significant benefit from COVID-19 vaccinations.

Our prior work has explored associations between lifestyle factors and several health outcomes (e.g., COVID-19 infections, Lyme Disease) [2,3]. Associated lifestyle factors and market segments with the relevant health outcomes (e.g., COVID-19) and healthcare decisions (e.g., vaccination rates) could potentially become an integral step for the implementation of coordinated vaccination interventions [2]. The ability to target a population in terms of its vaccination characteristics (e.g., vaccination acceptance, vaccination hesitancy) and its vaccination needs is becoming increasingly critical for the successful management and control of COVID-19. Vaccine hesitancy refers to the delay in acceptance or refusal of vaccines despite the availability of vaccine services [4]. This phenomenon is complex and context-specific, varying across time, place, and vaccines. It is influenced by factors, such as complacency, convenience, and confidence, which are all related to lifestyle attributes of individual households and communities—consumer behaviors, civic engagement, income, education, dietary preferences, and so forth [4,5].

We demonstrated the impact of lifestyle on COVID-19 vaccination by applying the CDC’s vaccination rates at the US county level. We utilized ESRI’s lifestyle segmentation system, which classifies all U.S. households into fourteen LifeModes categories, and identified segments that are associated with higher rates of vaccine hesitancy [6]. To effectively analyze the impact of vaccination on American households by their lifestyle characteristics, we specifically tested three research questions: “Which LifeModes have vaccination rates that are statistically higher/lower than average?”, “How has vaccine uptake/hesitancy changed over time among the different LifeModes?” and “How well can vaccination rates be predicted at sub-county levels using lifestyle segmentation data?” We focused on comparing each LifeMode’s mean to the national mean to ascertain spatial and temporal patterns of high-risk households and the effects of lifestyle on the vaccination rates in the United States.

## 2. Vaccine Hesitancy in the United States

In the United States, the COVID-19 vaccination rate has been influenced by vaccine hesitancy, which is related to a proportion of individuals who remain unwilling or uncertain about vaccination [4,5,6,7]. Psychographic, demographics, and geography play a major role in COVID-19 vaccine hesitancy, indicating a need for targeted messaging to high-hesitancy groups.

Vaccination acceptance is a proportion of people who have already received or are willing to receive the vaccine [4,5,6,7]. A national survey by Mejia and King assessed time trends and how each hesitancy group’s outlook changed regarding vaccination [4]. Analyzing the data by race, education, U.S. region, and Trump support in the 2020 election, they found the COVID-19 vaccine hesitancy was higher among young (ages 18–24), non-Asian people, and less educated (≤high school diploma) adults with a record of a positive COVID-19 test, not concerned about severity of COVID-19 and living in Donald Trump-elected regions.

Between January and May, vaccine hesitancy decreased in the population with a high school education or less but stayed constant with those who obtained higher levels of education. Although all racial groups observed a decrease in vaccination hesitancy in May, Black people and Pacific Islanders had the largest decrease, joining Hispanics and Asians. Higher hesitancy is observed among residents of counties with higher Trump support in the 2020 presidential election, whom neither trusted the vaccine nor the government. The difference in hesitancy between high and low Trump-elected counties increased over the period studied. Less hesitant groups were interested to wait and see if the vaccine was safe. Researchers also found that the lack of change with those with strong feelings about the vaccine were not likely to change easily and reaching that group with messages and incentives was crucial to increase the vaccine acceptance [8,9]. Researchers suggest targeted campaigns may have increased vaccine acceptance and access, since decreases in disparities by race and educational attainment were found.

King and Mejia investigated vaccine hesitancy by race and age subgroups. They found younger black people were more hesitant than younger white people in May, while the reverse was true in older populations [8,9]. This type of detailed analysis could help policymakers identify vaccine-resistant pockets. Survey participants had greater vaccine uptake with higher education levels compared to the general population. Due to this limitation, the percentage of people who refuse the COVID-19 vaccine is likely higher.

A cohort study by Siegler et al. [10] found that despite vaccine hesitancy having weakened between late 2020 and early 2021, inequities remained. Higher vaccine willingness converted into successfully delivered vaccinations; vaccine hesitancy converted into vaccine willingness (37%) and initial hesitancy of being vaccinated at follow-up (32%).

Calaghan’s study [11] analyzed the influence of demographics, political beliefs, and COVID-19 experiences on COVID-19 vaccine hesitancy by conducting a survey of 5009 American adults collected from 28 May–8 June 2020. A total of 31% percent of participants did not pursue getting vaccinated when the US began COVID-19 vaccinations. The two most mentioned reasons for vaccine refusal were concerns related to vaccine safety and effectiveness. Among sub-populations, women were most likely to be hesitant based on concerns about vaccine safety and efficacy. Black people were more likely to be hesitant than white people. In addition to concerns about safety and efficacy, black people lacked the needed financial resources or health insurance, and they had already contracted COVID-19.

In the context of the global vaccine distribution, vaccine nationalism presents a challenge to the equitable access to vaccines. Even though governments have the right and duty to ensure their citizens priority access to a COVID-19 vaccine, some governments will heavily prioritize the vaccine manufactured in their own country to their citizens [12]. The issue of equal access to vaccination led to equity-based initiatives such as the WHO-devised COVAX equitable vaccine access plan. By accelerating the development and manufacturing of vaccines as well as facilitating the equitable distribution of them, COVAX ensures said vaccines reach poor countries [13]. However, the success of the COVAX with respect to equitable access to the vaccine is likely to be limited, primarily because of the prevalence of vaccine nationalism [14].

We identified lifestyle segments that have a high propensity for vaccine uptake. High-risk lifestyle segments are clearly the areas where in the U.S. the public might benefit from getting vaccinated. Outreach and education campaigns targeted to these segments can reduce access and logistical barriers to vaccination. Based on their lifestyle preferences, these at-risk households could be patient communities in clinical trials’ campaigns. We mapped when and where each LifeMode had above/below the overall mean vaccination rate and how COVID-19 vaccination impacted different households. Our results recommend prevention and control policies to be implemented to those specific households.

## 3. Materials and Methods

### 3.1. Data

We integrated data from multiple sources into geographic information systems (GIS) to create a map exhibiting the concentration of the high-risk lifestyle segments (i.e., the tapestry is used to correlate lifestyle segments with COVID-19 vaccination rates.) As a first step, we examine what is ESRI’s tapestry segmentation and of what the tapestry dataset is comprised. Then, we provide information on the COVID -19 vaccination datasets we used for the analysis and further explain our study design and methods applied in detail.

#### 3.1.1. An Introduction to ESRI Tapestry Segmentation

We have chosen ESRI Tapestry data for our analysis because of ESRI’s authorized leadership as a location intelligence platform provider. According to the Forrester report, in their 30-criterion evaluation of location intelligence platform providers, ESRI is identified as one of the leaders among the nine most significant ones—ESRI, CARTO, Google, Hexagon, MapLarge, Microsoft, Oracle, Salesforce, and Syncsort [15]. Esri’s Tapestry Market Segmentation is also one of the most widely used geodemographic systems that provides an accurate, detailed description of American households and categorizes them into 67 distinctive consumer market segments based on socioeconomic, demographic, and psychographic data [16]. ESRI Tapestry segmentation system employs Experian’s Consumer View database [17], the Survey of the American Consumer from GfK MRI [18], and the U.S. Census American Community Survey [19]. ESRI Tapestry segmentation compiles these databases specifically to understand consumers’ lifestyle choices—what they buy and how they spend their free time, their beliefs, and life patterns based on geography, behavior, demographics, and psychographics—their income and demographic parameters such as race, gender, age groups, and marital status [19,20,21,22,23,24]. ESRI Tapestry helps gain insights into the markets, and which markets are being underserved, ultimately improving the performance of existing locations, finding optimal site locations, and investing resources wisely.

ESRI Tapestry segments are grouped into 14 LifeMode Groups with names such as “Rustic Outposts”, “Affluent Estates”, and “Family Landscapes”, which have commonalities based on lifestyle and life stages [16]. We downloaded ESRI Tapestry segmentation data from ESRI that contained the dominant LifeMode within each U.S. County and the number of households and percentages of the households in the county belonging to each LifeMode [25]. Appendix B Table A1 provides a detailed description of the LifeModes. A map of the dominant LifeMode for each U.S. County is provided in the Appendix A.

#### 3.1.2. COVID-19 Vaccination Rates

We downloaded COVID-19 vaccination rates from the CDC’s vaccination website which were updated daily on a county-by-county basis [22]. This dataset contains many variables, as there are many facets to vaccination status. For our analysis, the variable *series_complete_pct* was used [26]. This represents the percentage of people in the county considered “fully vaccinated”, meaning that they received two doses of the Pfizer or Moderna vaccine or one dose of the Johnson & Johnson vaccine. As the criteria for vaccine eligibility changed several times throughout 2021, we maintain temporal consistency by considering only the percentage of the entire population vaccinated, without when/who was eligible. Another consideration is that late in 2021, the FDA approved the distribution of booster doses for people more than six months removed from their most recent shot [27]. Our analysis does not account for boosters, as eligibility for the booster depends on the time at which the initial vaccination was received, making it difficult to compare across counties. Moreover, this study is principally focused on identifying populations who refuse the initial vaccination, so analyzing the booster uptake is not relevant to our goals. The CDC dataset also contains raw counts of vaccinated people as well as population data by county. These two variables are also utilized for our analysis. Figure 1 shows the percent that is fully vaccinated by county on 1 January 2022.

On 1 January 2022, 58.5% of the U.S. population was considered fully vaccinated. However, the mean vaccination percentage for all U.S. counties (not weighted for population) on 1 January 2022 was 49%. This indicates that counties with higher population (urban) disproportionately had higher vaccination rates, while counties with low population (rural) disproportionately had lower vaccination rates. Vaccination rates appeared to be normally distributed. A histogram of county-by-county vaccination rates is provided in the Appendix A. The maximum vaccination rate of 95% was found in Santa Cruz County, Arizona, where most of the population is elderly. Vaccine uptake appears to be highest in the urban corridors of the Northeast, the West Coast, and in the retirement communities of Arizona and Florida, and lowest is in the rural areas of the Great Plains and the Southeast.

### 3.2. Methods

This study is a population-based observational study that explores associations between vaccination uptake (i.e., high/low vaccine uptake) and LifeModes. We further describe methods applied in detail.

#### 3.2.1. Temporal Analysis of High and Low Vaccine Uptake LifeModes

This section of the methods is devoted to discovering which LifeModes are associated with statistically high/low vaccine uptake. Counties are grouped by their dominant LifeMode, and the mean vaccination rates for each LifeMode are calculated. A 95% confidence interval for the mean vaccination rate for each LifeMode is generated using bootstrapping with 1000 replications. The limits of this interval can be compared to the nationwide statistics to determine whether each LifeMode’s vaccine uptake differs significantly from the nationwide average. The process is repeated using the median instead of the mean. If a LifeMode’s mean vaccination rate confidence interval lies completely above the nationwide mean and the median vaccination rate confidence interval lies completely above the nationwide median, that LifeMode is designated as a “High” vaccination status. If a LifeMode’s mean vaccination rate confidence interval lies completely below the nationwide mean and the median vaccination rate confidence interval lies completely below the nationwide median, that LifeMode is designated as a “Low” vaccination status. If neither of the above criteria are met, the LifeMode is designated as “Neutral”. Appendix A presents mean vaccination rate confidence interval and Appendix A represents median vaccination rate confidence interval in Appendix A. 

This analysis is repeated quarterly, using vaccination rates obtained from 1 July 2021; 1 October 2021; and 1 January 2022. This enables the comparison of vaccine uptake among LifeModes across different temporal periods. July 2021 is chosen as the first quarterly date to examine because it is the first quarter in which all American adults would have had both access to the vaccines and sufficient time to reach “fully vaccinated” status. Earlier in 2021, vaccines were only available to the elderly and those with vulnerability due to underlying health conditions.

#### 3.2.2. Principal Component Analysis

The next component of our study is the prediction of vaccination rates using ESRI’s tapestry segmentation and other demographic variables. To reduce the complexity of the dataset, a principal component analysis is applied prior to modeling. The following variables were included in the analysis: the percentage of households in the county belonging to each LifeMode (14 variables), the percentage of non-white persons, the percentage with a bachelor’s degree or higher, the median income, the median age, and the percentage of the vote received by Donald Trump in the 2020 presidential election. The demographic variables are all obtained from the 2015–2019 American Community Survey (ACS) [28], except for the voting data, which was downloaded from the GitHub under US County Election results (data originally obtained from the New York Times) [29]. These specific demographic variables were chosen because the King et al. study [8,9] indicated that they were effective predictors of vaccination rates. This resulted in a total of 19 variables aggregated at the county level. The PCA generated 19 principal components. The percentage of variance explained by each of the first ten components is displayed in the Appendix A. The first seven components were chosen for use in the model as they all had an eigenvalue greater than 1 and there was a reasonable cutoff in variance explained between components 7 and 8. Therefore, components 1 through 7 were exported to a new matrix and used as the predictor variables in the logistic regression analysis.

#### 3.2.3. Logistic Regression Analysis

Regression analysis is used to model vaccination rates using lifestyle segments as predictor variables. The resulting model may then be leveraged to predict vaccination rates at sub-county levels (e.g., census tracts, block groups) based on the lifestyles found there. Because the response variable (vaccination rates) represents a binary choice (fully vaccinated or not fully vaccinated), logistic regression is the appropriate model.

The assumptions for logistic regression [30] are mostly satisfied, albeit a few concerns. The response variable is binary. There is no multicollinearity among the predictors since they come from principal components. The sample size is sufficiently large. Unfortunately, there is the possibility of some spatial autocorrelation in the observations. This will be examined using mapping software after the model is completed. There are also some outliers in the data, but these tend to be the very populous and diverse counties and there is no reason to believe this is due to incorrect reporting. Thus, they are left in the data.

It was also noted from the maps in Figure 1 that there seem to be systematic issues with the reporting of vaccine percentages in Georgia, as the vaccination rates within the state are unusually low and the area of unusually low rates correlates perfectly with the state boundary. Therefore, the Georgia counties are discarded from the model. These issues were not present in the predictor variables (demographic and LifeMode data) so the model can still be used to make predictions for Georgia counties.

To choose the best fit, the 10-fold cross-validation method is used. The final model is chosen according to the initial model with the lowest RMSE. There was a significant degree of spatial autocorrelation [31] in the model residuals. Specifically, the data was spatially clustered (Moran’s I = 0.336, z = 56.9, *p* < 0.001). Using a non-parametric modeling method can reduce the problems associated with this autocorrelation. Therefore, a random forest model is also generated, and its effectiveness is compared with the logistic regression model.

#### 3.2.4. Random Forest

Due to the issues associated with this dataset regarding logistic regression (i.e., spatial autocorrelation, outliers), a nonparametric modeling method is also attempted. The random forest model was the nonparametric method chosen for this analysis [32]. This model consists of building a collection of decision trees, with each individual tree based on a subset of the variables in the dataset and a random sample of the observations obtained through bootstrapping. In our case, 1000 trees will be built, each containing 5 of the 19 possible variables. In a random forest model, the predictor variables need not be normally distributed or uncorrelated. This means that we can return to using the original variables, not the principal components. There are 19 predictor variables: the percentage of households in the county belonging to each LifeMode (14 variables), the percentage of nonwhite persons, the percentage with a bachelor’s degree or higher, the median income, the median age, and the percentage of the vote received by Donald Trump in the 2020 presidential election. The response variable is the percentage of the entire population considered fully vaccinated. Once again, the observations from Georgia are not included in the analysis due to an apparent bias.

## 4. Results

### 4.1. Temporal Analysis of High and Low Vaccine Uptake LifeModes

We compared the mean vaccination rates for each LifeMode to the nationwide mean. Several clusters become apparent. LifeModes 1 (Affluent Estates), 2 (Upscale Avenues), 3 (Uptown Individuals), and 13 (Next Wave) seem to have the highest vaccination rates. A middle cluster includes LifeModes 4 (Family Landscapes), 5 (GenXUrban), 6 (Cozy Country Living), 7 (Sprouting Explorers), 8 (Middle Ground), 9 (Senior Styles), 11 (Midtown Singles), 12 (Hometown), and 14 (Scholars and Patriots). Finally, LifeMode 10 (Rustic Outposts) has the lowest vaccination rates. Figures of 95% confidence intervals for both the mean and median vaccination rate on 1 January 2022, for each of the fourteen LifeModes, are provided in the Appendix A. The medians show roughly the same pattern: three clusters of LifeModes. Most LifeModes have a mean *below* the nationwide mean, while most LifeModes have a median *above* the nationwide median. Thus, a LifeMode is only awarded high vaccination status if its mean and median are both above the respective nationwide mean and median. This results in LifeModes 1 (Affluent Estates), 2 (Upscale Avenues), and 13 (Next Wave) receiving high vaccination status. Similarly, a LifeMode is awarded low vaccination status if its mean and median are both below the respective nationwide mean and median. The only LifeMode meeting these criteria is 10 (Rustic Outposts). The remainder of the LifeModes are designated “neutral” about their vaccination status. The same process is then repeated for data from 1 July 2021, and 1 October 2021 to see if the vaccination propensity of the various LifeModes might have changed over time. Table 1 displays the vaccination status of each LifeMode on each date:

Table 1 reveals that only one change in vaccination status occurred between the July 2021 and January 2022 data: Upscale Avenues started out as neutral in July and shifted to high in October and January. Affluent Estates and Next Wave remained consistently high across the time period; Rustic Outposts remained consistently low. Figure 2 yields more insight by looking at the precise changes in the mean vaccination rates over time.

For all of the groups, the vaccination rate increased more between July and October than between October and January. This is to be expected because there were less unvaccinated people remaining in the second period compared to the first. Between July and October, Uptown Individuals saw the largest increase in vaccination rate (+17.6%), while Cozy Country Living (+7.0%) had the smallest increase (nationwide: +9.5%). Between October and January, Next Wave (+7.7%) had the largest increase in vaccination rate, while Cozy Country Living (+4.7%) once again had the smallest increase (nationwide: +6.1%). Figure 3 provides a nationwide map showing the locations of the LifeModes with high, low, and neutral vaccination status. By mapping the LifeModes according to their vaccination status, the neighborhoods where vaccine hesitancy is high can be predicted and interventions can be considered.

### 4.2. Logistic Regression Analysis

Table 2 shows the coefficients and significance levels for each of the seven predictor variables. Table 3 shows the measures of fit for the model.

The first six predictors (PC1 through PC6) are significant in the model; PC7 is not. The model’s performance is not impressive with an R-squared of 0.401. The maps in Figure 4 and Figure 5 show the model predictions by county and the residuals. Some spatial autocorrelation (clustering) in the residuals is clear and may be the reason for the low R-squared value.

Figure 5 indicates some spatial clustering in the model residuals. There is a cluster of positive residuals (modeled vaccination rates higher than reality) in areas from the northern Great Plains; these are extremely isolated rural areas that are mostly white and politically conservative. Because of the sparse, isolated nature of the population there, limiting the opportunities for viral spread, residents may feel that COVID-19 mitigation is not a priority in their lives. Several clusters of negative residuals (real vaccination rates higher than modeled) are also apparent. One, in the Southwest, consist of rural counties which are inhabited mostly by Native Americans (for the AZ/NM counties) or Hispanics (for the counties along the TX border). What we are seeing here appears to be racial/ethnic minorities adopting the vaccine at higher rates than would be expected based on their lifestyle classifications. Additionally, looking closely at other parts of the West such as the Dakotas and Montana, counties which contain Native American reservations have higher than expected vaccination rates. Altogether, it appears that the model is not accounting for race/ethnicity to the extent that it should, which is surprising because the model did contain minority population as a variable. Perhaps in the calculation of the principal components, the aspect of minority population is being overshadowed by the variations of lifestyle types. The random forest model, which is not based on the principal components, may provide more accurate results.

### 4.3. Random Forest Analysis

The random forest model slightly outperformed the logistic regression model, yielding 59.7% of variance explained (logistic: 40.1%). The residual standard error was 7.23% (logistic: 8.84%). Figure 6 (below) provides insight into what variables were most important in the model. The most important variable by far was Republican voting habits, with Republican counties less likely to take the vaccine. The other variables in order of importance were: College Educated, Rustic Outposts, Minority Race/Ethnicity, Median Income, Median Age, Upscale Avenues, Cozy Country Living, Affluent Estates, Senior Styles, GenXUrban, Hometown, Middle Ground, Family Landscapes, Uptown Individuals, Midtown Singles, Sprouting Explorers, Next Wave, and Scholars and Patriots.

Figure 7 and Figure 8 show the random forest resultant maps and convey the effectiveness of the random forest model compared to the logistic regression model. Figure 8 (random forest residuals) corresponds exactly with Figure 5 (logistic regression residuals) using identical color schemes. While Figure 4 shows that the logistic regression model had numerous counties, which suffered an error of more than 20%, the random forest model (Figure 7) had no such counties. The random forest model produced an error of less than 10% for the overwhelming majority of counties. Counties with errors of greater than 10% are few and far between and are all rural and sparsely populated. The model still underestimates vaccination rates in several western counties with Native American reservations, but these are the only places where such errors are observed. Based on these results, it is clear that the random forest model should be preferred for the modeling of vaccination rates.

## 5. Discussion

Ensuring access to COVID-19 vaccines and improving vaccination rates for at-risk populations can help address the disparate health effects of the virus [1]. The common approach by the public health officials is to educate people about why vaccination is a good idea [33]. However, there is more nuance to vaccine hesitancy than we realize [4,5,6,7,8,9]. The formation of attitudes about vaccination is complex. In this paper, we describe how aspects of the human behavioral context, such as lifestyles, led to geographically targeted at-risk populations based on their lifestyle traits. By mapping the LifeModes according to their vaccination status, we can reach the most hesitant subgroup of Americans and predict the households where vaccine hesitancy is high, and interventions can be considered.

Mapping the LifeModes according to their vaccination status is the cornerstone of public health policy and interventions to promote vaccination. To reach vaccination goals for everyone in the community, healthcare providers, public health officials, and immunization partners should target their communications towards these households and consider their lifestyle preferences when tailoring messages and strategies to increase vaccination acceptance and uptake. Understanding the current and past vaccination trends over the last several months is important for projecting what may happen in the next couple months and also help policymakers identify vaccine-resistant pockets. The wide disparity in the acceptance of the vaccines decreased over time. Getting enough people vaccinated can reduce the speed of new variants spreading as well.

Our findings demonstrate that vaccine uptake appears to be highest in the urban corridors of the Northeast, the West Coast, and in the retirement communities of Arizona and Florida, and lowest in the rural areas of the Great Plains and Southeast. Looking closely at other parts of the West such as the Dakotas and Montana, counties which contain Native American reservations have higher vaccination rates. Racial/ethnic minorities also adopt the vaccine at higher rates. The most effective predictor of vaccination hesitancy was Republican voting habits, with Republican counties less likely to take the vaccine. This finding really highlights the politicization of public health recommendations. The other predictors in order of importance were college education, minority race/ethnicity, median income, and median age. The following LifeModes were associated with high vaccination rates consistently above national average: Upscale Avenues, Next Wave, Affluent Estates, and Uptown Individuals. Residents of these LifeModes are affluent and well educated. Affluent Estates and Upscale Avenues represent health conscious and early adapter households of new products and technology, so this explains their high vaccination rates. GenXUrban, Middle Ground, Sprouting Explorers, Senior Styles, Scholars and Patriots closely follow nationwide vaccination rates, but are lower than the national average. There is a consistent increase in the lower rate LifeModes, such as Hometown, Family Landscapes, and Midtown Singles. Cozy Country living exhibits the smallest increase. Rustic Outposts has the lowest vaccination rate.

Having an effective marketing strategy based on at-risk households’ lifestyle preferences could lead to higher vaccination rates. For example, Family Landscape are successful young families, residing in suburban and semirural areas and are more likely to vaccinate than individuals without children. It could be that parents, with their regularly scheduled doctor’s visits, children’s immunization schedules, and concern about their child’s welfare, are also more engaged in their own immunization health.

Meanwhile, Rustic Outposts represents the LifeMode with the lowest vaccination rate and it represents mostly poor, rural, southern households. This lifestyle is also heavily Republican in political preference, so to increase vaccine uptake in these communities, the most important step is to reduce the politicization of the vaccine. Specific strategies for reducing the politicization of the vaccine are outside the bounds of our expertise and thus beyond the scope of this paper but would certainly be a worthwhile pursuit for political scientists.

The below average vaccination adherence of the Midtown Singles and Scholars and Patriots LifeModes is also surprising from a political perspective, as these LifeModes tend to consist of younger people with more progressive political views. However, the young median age of the population is probably the key to understanding this phenomenon. Age was a significant factor in the random forest model, as older populations were more likely to be vaccinated than younger populations. The young median age associated with these LifeModes likely explains the lower-than-expected vaccination rates. Hesitancy among these groups could likely be reduced by education about the importance of vaccination in reducing community transition, urging young people to obtain the vaccination for this purpose even if the disease itself poses a relatively small threat to these population groups.

This lifestyle segmentation not only helps to geographically target at-risk populations, but also to conduct more effective intervention, prevention, and treatment by increasing vaccine confidence in at-risk households based on their lifestyle traits. For example, Rural Outposts exhibit a lack of trust in government and science and consequently in vaccines. Public health messages with better assistance in potential side effects and safety need to reach out to Rural Outposts households to increase vaccine confidence.

Improvement activities can possibly be implemented to increase adhesion in these at-risk communities where people live, work, learn, pray, play, and gather. Our ability to map high-risk lifestyle segments for geographic areas smaller than the county level allows for more targeted interventions and activities. Our findings could help public health policy makers narrow their focus on the most promising groups or lifestyle segments by households, zip codes, and block groups [2]. This finer scale information could be used to improve vaccine delivery and operations that can effectively achieve the target populations. Staff and volunteers can be recruited to talk with people about vaccinations for themselves and their loved ones, answer their vaccine questions, and schedule their vaccination appointments. Vaccination events can be promoted, and vaccination information can be distributed to teachers, parents, and students through various communication means such as flyers, emails, and newsletters [34]. Employers can be encouraged to help employees get vaccinated. Education and outreach efforts and pop-up vaccination clinics can be coordinated with faith-based organizations, community centers, recreation centers, and local parks. Geo-fencing and hyper-target digital marketing strategies through social media and virtual events can be implemented to increase vaccination uptake in these high-risk lifestyle segments [34].

The locations of high-risk LifeModes are clearly the areas in the U.S. where clinical trials of new vaccines for COVID-19 variants can recruit patients and advertise with the right message based on their lifestyle preferences and determine their overall motivation to engage in clinical research.

We also found that the effect of proximity to highly infected areas play a role in vaccination status and vaccine awareness. Educational tools, clinical trials campaigns, and public health messages could be issued to these at-risk lifestyle households in those areas where COVID-19 infections are high. For example, rural counties in the Southwest that are inhabited mostly by Native Americans (for the AZ/NM counties) or Hispanics (for the counties along the TX border) adopt the vaccine at higher rates than would be expected based on their lifestyle classifications. Additionally, looking closely at other parts of the West, such as the Dakotas and Montana, counties which contain Native American reservations have higher than expected vaccination rates. Contrarywise, because of the sparse, isolated nature of the population in areas from the northern Great Plains, limiting the opportunities for viral spread, residents may feel that COVID-19 mitigation is not a priority in their lives. In other words: low-trust individuals who live close to an affected area harbor more favorable views about vaccination for COVID-19. This implies that citizens who are skeptical of the CDC and similar institutions base their vaccination decision-making to some degree on whether a given disease occurs in close vicinity to their community.

The main limitations with our findings come from inherent difficulties with the vaccination adherence data. These data are reported on a state-by-state basis, so uniformity in counting methods across states is not guaranteed. Moreover, states do not provide complete details on their counting methodologies, so the exact reasons for these discrepancies cannot be uncovered. For example, as discussed earlier, the entire state of Georgia experienced unusually low vaccination rates compared to the nationwide average and even the immediately neighboring states. This was likely due to some aspect of Georgia’s data collection process, introducing a negative bias. We remedied this by excluding Georgia’s data from the model. While there may be discrepancies among other states, no other states demonstrated an obvious bias compared to the nation, and therefore no other states were removed.

## 6. Conclusions

We conclude that we can identify patterns of vaccination uptake and hesitancy by correlating COVID-19 vaccination data with lifestyle segments. We can predict the neighborhoods where vaccine hesitancy is high, and interventions can be considered. Linking segments to geographically identified patients (e.g., vaccinated, vaccine hesitant) could support the ability to estimate vaccination acceptance and uptake, help policymakers identify vaccine-resistant pockets, and predict demand for health care delivery. These associations can lead to more efficient, more targeted, and more cost-effective vaccinations [2]. Our methodology for COVID-19 vaccination in this paper, and in our previous papers [2,3], is a model for transforming lifestyles into the incidence of diseases, along with other health metrics such as vaccination.

Market Intelligence tools (i.e., segmentation tools) [2] can assist researchers and key health care stakeholders in exploring the causal pathways between drivers (e.g., lifestyle traits) and outcomes (e.g., vaccination rates). Our ability to profile a target population of LifeModes and lifestyle segments associated with vaccination hesitancy and their lifestyle traits can help inform continued efforts to promote vaccination uptake, improve vaccination rates, mitigate the harms from COVID-19, and ultimately lower hospital readmission rates related to COVID-19 and its underlying medical conditions.

Our findings illustrate the power of leveraging market intelligence tools to improve health outcomes and health care delivery. Health care delivery organizations of all kinds, from independent individual provider units to large integrated health systems and public health policy makers, are in need to market and distribute vaccines to these households and communities. By designating at-risk market segments down to the Census block group level by psychographics, demographics, and socioeconomics, they can better reach and engage their patients and tailor their policies and strategies in which patients are communicated with differently depending on their personalities and preferences.

There are a number of gaps in our knowledge around health consumer technology in research that follows from our findings and would benefit from further research, including patient technology advancements that were made during the pandemic and resulted in tremendous technological growth that meets the standards of care that patients want. Part of that involves a care plan that encompasses patient communication between themselves and their provider, remote monitoring, and check-ins on additional aspects of their care plan, such as clinical trials and vaccination status. Future research could explore the leadership role of market intelligence tools intertwined with patient communication. More methodological work is needed on how to robustly capture the morbidity data, maybe through claims data and health metrics (e.g., vaccination data), or a robust analysis and exploration of innovative uses of GIS in visualizing and integrating health data and electronic health information to better target populations or geographic areas in the most need of public health interventions, ultimately improving population health outcomes. It would be helpful to further explore applying our concept with a real time segmentation system that does not rely on survey data, which captures the intent but real time behavior with the live aspects of social media, web behavior, and real-world visitation.

## Figures and Tables

**Figure 1 ijerph-19-07732-f001:**
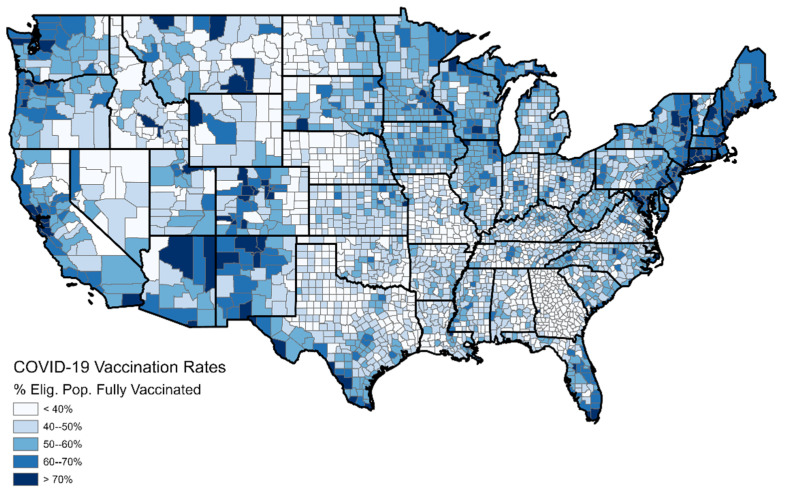
County–by–county vaccination rates for 1 January 2022.

**Figure 2 ijerph-19-07732-f002:**
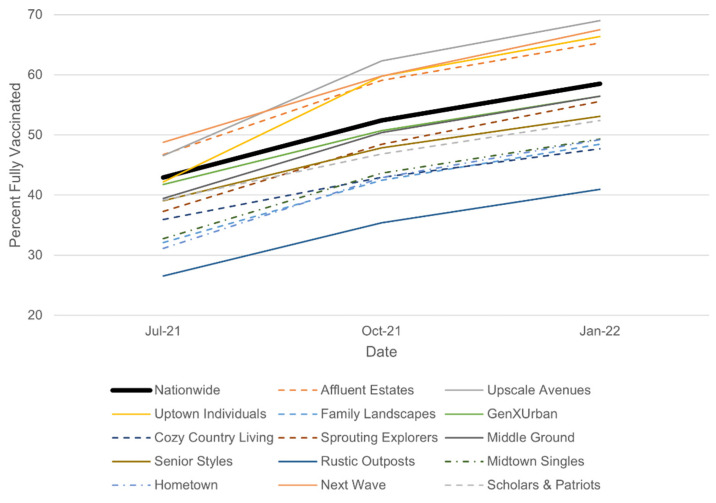
Mean vaccination rates for each of the fourteen LifeModes vs. the nationwide mean plotted over time.

**Figure 3 ijerph-19-07732-f003:**
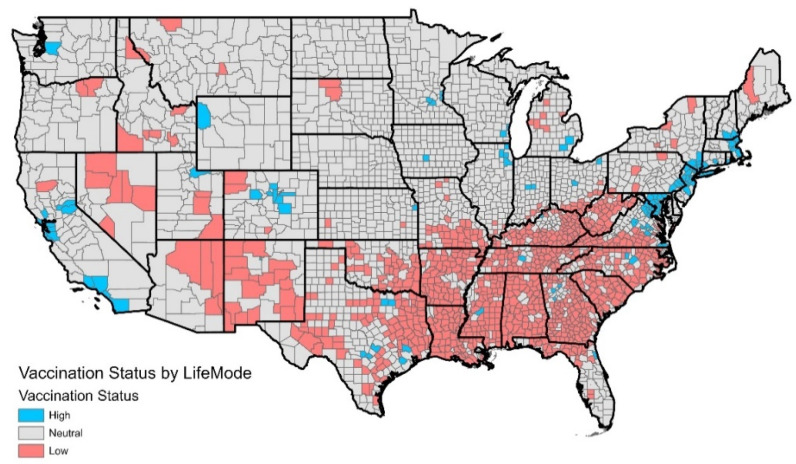
Vaccination status by county across the United States, estimated using the dominant LifeMode assigned to each county.

**Figure 4 ijerph-19-07732-f004:**
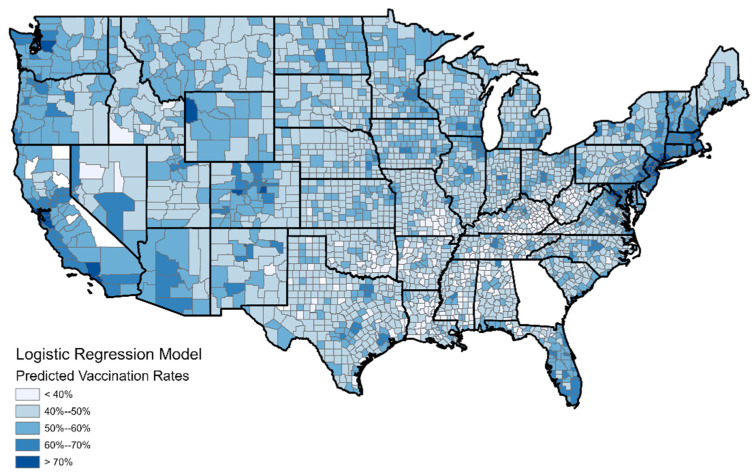
Predicted vaccination rates by county according to logistic regression modeling.

**Figure 5 ijerph-19-07732-f005:**
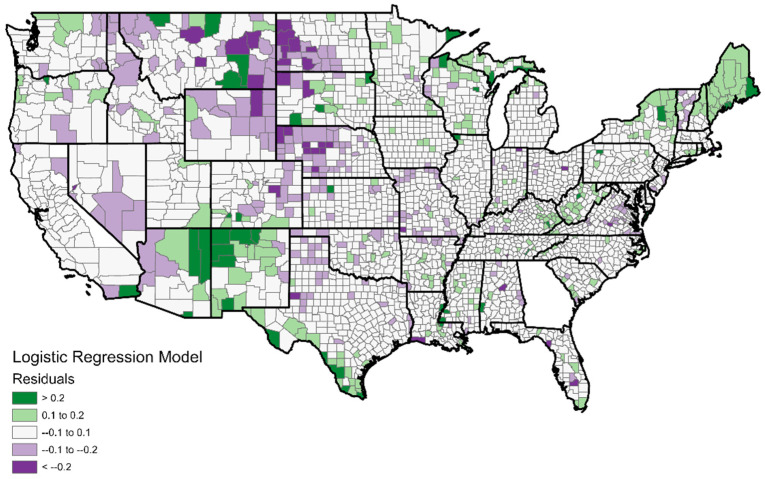
Residuals from logistic regression model. Green indicates vaccination rates were higher than expected; purple indicates vaccination rates were lower than expected.

**Figure 6 ijerph-19-07732-f006:**
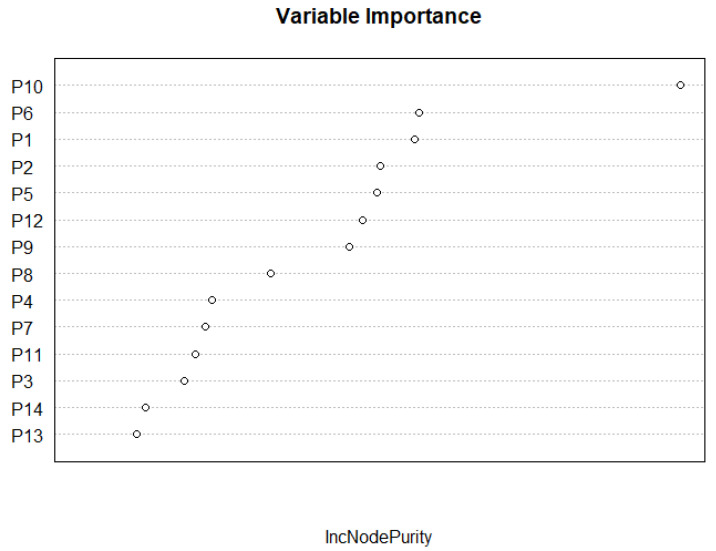
Variable Importance in the random forest model.

**Figure 7 ijerph-19-07732-f007:**
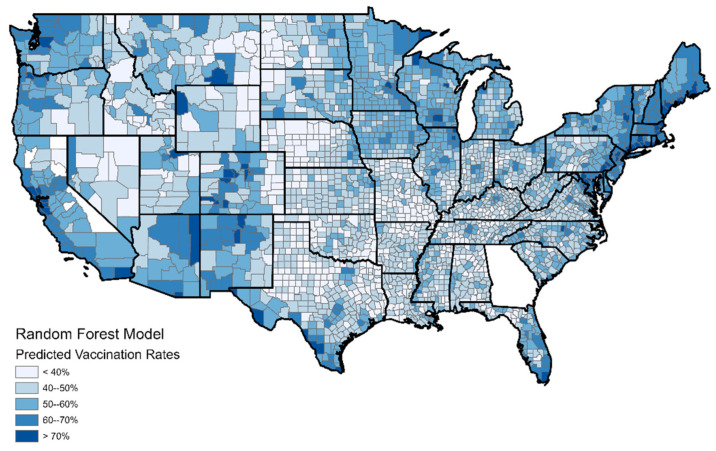
Predicted vaccination rates by county according to the random forest model.

**Figure 8 ijerph-19-07732-f008:**
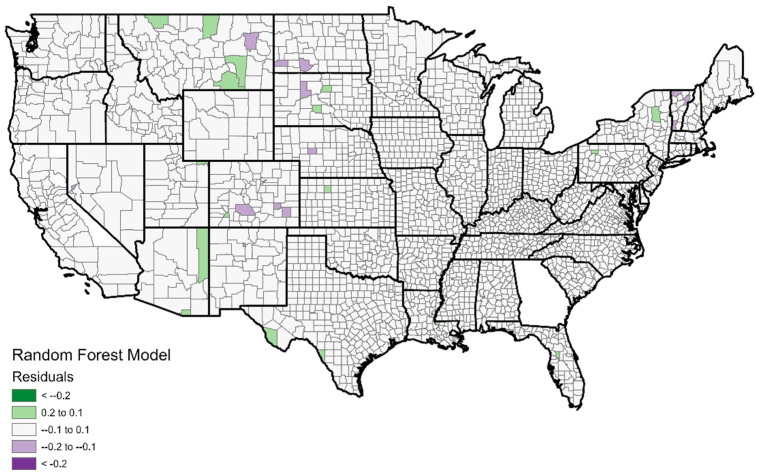
Residuals from random forest model. Green indicates vaccination rates were higher than expected; purple indicates vaccination rates were lower than expected. Note that the predictions are much more accurate in general than those of the logistic regression model.

**Table 1 ijerph-19-07732-t001:** Vaccination status for each LifeMode on 1 July 2021; 1 October 2021; and 1 January 2022.

LifeMode	1 July 2021	1 October 2021	1 January 2022
(1) Affluent Estates	High	High	High
(2) Upscale Avenues	Neutral	High	High
(3) Uptown Individuals	Neutral	Neutral	Neutral
(4) Family Landscapes	Neutral	Neutral	Neutral
(5) GenXUrban	Neutral	Neutral	Neutral
(6) Cozy Country Living	Neutral	Neutral	Neutral
(7) Sprouting Explorers	Neutral	Neutral	Neutral
(8) Middle Ground	Neutral	Neutral	Neutral
(9) Senior Styles	Neutral	Neutral	Neutral
(10) Rustic Outposts	Low	Low	Low
(11) Midtown Singles	Neutral	Neutral	Neutral
(12) Hometown	Neutral	Neutral	Neutral
(13) Next Wave	Neutral	Neutral	Neutral
(14) Scholars and Patriots	Neutral	Neutral	Neutral

**Table 2 ijerph-19-07732-t002:** Model coefficients and significance levels for the predictor variables (principal components 1 through 7).

	Estimate	Z-Value	*p*-Value
Intercept	−0.05318	−258	<0.001 *
PC1	0.1263	2150	<0.001 *
PC2	0.07037	591	<0.001 *
PC3	−0.0154	−198	<0.001 *
PC4	−0.06104	−672	<0.001 *
PC5	0.06659	627	<0.001 *
PC6	0.01086	94	<0.001 *
PC7	−0.00005	−0.378	0.705

* Indicates Significance.

**Table 3 ijerph-19-07732-t003:** Logistic regression model measures of fit.

RMSE	8.84%
R-Squared	0.401
MAE	0.0652

## Data Availability

Raw data “Vaccination rates” were derived from the CDC Wonder database available in the public domain at https://data.cdc.gov/Vaccinations/COVID-19-Vaccinations-in-the-United-States-County/8xkx-amqh (accessed on 1 February 2022). The authors confirm that the data supporting the findings of this study are available within the article and its Appendix A. The GIS data and maps of High and Low Risk LifeModes are not publicly available due to the commercialization of research findings.

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
