# Peer review of "The Effects of Lifestyle on COVID-19 Vaccine Hesitancy in the United States: An Analysis of Market Segmentation"

_ijerph, 2022, doi:10.3390/ijerph19137732_

Round 1

Reviewer 1 Report

Q1. Introduction should be shortened.

Q2. Methods should be shortened

Q3. In discussion strengths and limits section is lacking

Q4. In discussion/conclusion explain how the results of your research can improve public health policies and strategies

Q5. Report the study design in methods section

Author Response

Authors’ Response

 We thank you for your suggestions. Please find below in bold are our answers to your questions.

Response to Reviewer 1:

Q1. Introduction should be shortened.

We have significantly shortened introduction into three paragraphs.

Q2. Methods should be shortened

We have shortened methods as much as we can though other reviewers wanted us to explain ESRI tapestry in more detail. So you will find we added more to that section under methods.

Q3. In discussion strengths and limits section is lacking

We have explained the strengths and limitations of the study better and added a paragraph on how the results of our research can improve public health policies and strategies. We also added a paragraph on the main limitations with our findings in discussions section.

Q4. In discussion/conclusion explain how the results of your research can improve public health policies and strategies

In discussion/conclusion, we have explained how the results of our research can improve public health policies and strategies.

Q5. Report the study design in methods section

We have reported the study design in methods sections.

Reviewer 2 Report

I have read the article that I find interesting, I think it could be important for readers internationally and should be published. However, I believe we can improve some aspects that I suggest:

- I believe that in the initial part it is possible to underline the usefulness of vaccination and that efforts have been made in the world to carry out global vaccination activities even in countries that are not suitable for this point of view.For example, you can mention:

doi: 10.3390 / vaccines9060538.

 doi: 10.1186/s12992-021-00763-8.

 doi: 10.1111/1468-0009.12503.

- the limitations of the study should be better expressed and whether further research is possible in the future

- it could be suggested if improvement activities are possible to increase adhesion

the authors should better explain what the "ESRI Tapestry Segmentation" system used is and what kind of authority and investigative capacity has.
- Furthermore, the authors also make very direct statements in the discussion, for example they indicate that due to the Republican vote there is poor adherence to the vaccine. Do we have any other evidence to support these claims?
Could there be confounding factors behind this data? are they uniform across all states or are there variations? These aspects should be better presented.

- The authors often use the expression of herd immunity, however it does not seem possible to achieve this goal with COVID 19 due to its characteristics of transmissibility despite vaccination. So these experiences should be removed

Author Response

Authors’ Response

 We thank you for your suggestions. Please find below in bold are our answers to your questions.

 Response to Reviewer 2:

I have read the article that I find interesting, I think it could be important for readers internationally and should be published. However, I believe we can improve some aspects that I suggest:

- I believe that in the initial part it is possible to underline the usefulness of vaccination and that efforts have been made in the world to carry out global vaccination activities even in countries that are not suitable for this point of view. For example, you can mention:

doi: 10.3390 / vaccines9060538.

 doi: 10.1186/s12992-021-00763-8.

 doi: 10.1111/1468-0009.12503.

We have mentioned all the suggested articles and cited them in text and included in references #12, #13, #14

- the limitations of the study should be better expressed and whether further research is possible in the future

We have added a paragraph on the limitations of the study in discussions and future possible research in conclusions.

- it could be suggested if improvement activities are possible to increase adhesion

We have included a paragraph in discussions on improvement activities to increase adhesion.

the authors should better explain what the "ESRI Tapestry Segmentation" system used is and what kind of authority and investigative capacity has.

We explained ESRI tapestry segmentation system more in detail and cited Forrester report endorsing Esri’s investigative and authoritative capacity.

- Furthermore, the authors also make very direct statements in the discussion, for example they indicate that due to the Republican vote there is poor adherence to the vaccine. Do we have any other evidence to support these claims?

This result comes from our random forest model, which showed that voting patterns were significantly associated with vaccination adherence. This was in fact the most influential variable in the final model.

Could there be confounding factors behind this data? are they uniform across all states or are there variations? These aspects should be better presented.

There is the possibility of differences in data collection between different states, although it is impossible to be aware of all the different ways that data is collected in the respective states. States with unusual, bias-introducing data collection practices became evident when looking at the model residuals (e.g., see our discussion of Georgia) and were therefore removed from the analysis. Added a paragraph in discussion to address this.

- The authors often use the expression of herd immunity; however, it does not seem possible to achieve this goal with COVID 19 due to its characteristics of transmissibility despite vaccination. So these experiences should be removed

We have removed the expressions related to herd immunity in the discussion and conclusion sections and else it was used.

Reviewer 3 Report

The paper provides a very interesting analysis of factors affecting anti COVID19 vaccine propensity

in the United State population. The authors compared the CDC’s vaccination rates to lifestyle segmentation to identify the segments with the highest vaccine hesitancy. 

·      In general the study shows an increase of vaccination rate between July 2021 and January 2022, but only the “upscale avenues” group change its propensity from neutral to high. How can be this change explained?

·      The politicization of vaccine recommendation justifies the low vaccination rates in Republican counties, but how can it be overcome?

·      The relative low adherence to vaccination in scholar and patriots and in midtown singles is quite surprising. What strategy can be applied to reduce their hesitancy?

·      Please, check figure numbers in page 9 and 10. 

Author Response

Authors’ Response

We thank you for your suggestions. Please find below in bold are our answers to your questions.

Response to Reviewer 3:

 In general, the study shows an increase of vaccination rate between July 2021 and January 2022, but only the “upscale avenues” group change its propensity from neutral to high. How can be this change explained?

       LifeModes are categorized by their vaccination percentages relative to the national average, not their absolute vaccination percentages. While most LifeModes saw their absolute vaccination rates increase between July 2021 and January 2022, the nationwide rate would have also increased, so changes in the relative vaccination rates are not guaranteed.

       The politicization of vaccine recommendation justifies the low vaccination rates in Republican counties, but how can it be overcome?

       This is probably beyond the scope of our study and our expertise. Added a sentence in the Discussion section to note this.

 The relative low adherence to vaccination in scholar and patriots and in midtown singles is quite surprising. What strategy can be applied to reduce their hesitancy?

       Vaccination adherence likely lags in these groups due to their younger population makeup. A paragraph was added to discuss this.

Please, check figure numbers in page 9 and 10.

       These were fixed, thank you for catching this.

Round 2

Reviewer 2 Report

I believe that the work has improved a lot and that the authors have listened to the suggestions. In my opinion the article is publishable